# New Practices in Transcatheter Aortic Valve Implantation: How I Do It in 2023

**DOI:** 10.3390/jcm12041342

**Published:** 2023-02-08

**Authors:** Ana Paula Tagliari, Maurizio Taramasso

**Affiliations:** 1Cardiovascular Surgery Department, Hospital São Lucas da PUC-RS, Porto Alegre 90619-900, Brazil; 2Cardiovascular Surgery Department, Hospital Mãe de Deus, Porto Alegre 90880-0481, Brazil; 3HerzZentrum Hirslanden Zurich, Clinic of Cardiac Surgery, 8008 Zurich, Switzerland

**Keywords:** transcatheter aortic valve implantation, transcatheter aortic valve replacement, trends, minimalist approach, conscious sedation

## Abstract

Transcatheter aortic valve implantation (TAVI) went through a huge evolution in the last decades. Previously performed under general anesthesia, with transoperative transesophageal echocardiography guidance and using cutdown femoral artery access, the procedure has now evolved into a minimalist approach, with local anesthesia, conscious sedation, and the avoidance of invasive lines becoming the new standards. Here, we discuss the minimalist TAVI approach and how we incorporate it into our current clinical practice.

## 1. Introduction

Since the first transcatheter aortic valve implantation (TAVI) in 2002 [1], the procedure has evolved in terms of device technology, procedural indications, and operative techniques. One of the most evident changes has been a trend towards a more minimalist TAVI approach.

The key principles for a successful minimalist TAVI include optimizing all pre-, peri-, and post-procedural steps of patient care; avoiding general anesthesia (GA); minimizing procedural sedation; standardizing peri-operative management; shortening the procedural time; and prioritizing early hospital discharge. This approach has been shown to be safe and capable of reducing the overall hospital costs, without increasing adverse outcomes [2].

However, there is still a lack of standardization about what the minimalist TAVI approach means. A variety of protocols have been proposed without a clear consensus on specific components or patient eligibility criteria. This variability leads to heterogeneous studies and makes it difficult to evaluate outcomes.

This article provides an up-to-date review of the minimalist TAVI approach, the main protocols suggested by experts, and the outcomes achieved when minimalistic strategies are adopted. Lastly, we discuss how we have incorporated a minimalist approach into our daily clinical practice.

## 2. The Minimalist TAVI Approach

Minimalist TAVI protocols, in generals, propose replacing GA by conscious sedation (CS) and/or local anesthesia (LA), preferring transthoracic echocardiography (TTE) assessment instead of transoperative transesophageal echocardiography (TOE), and replacing cutdown femoral artery access by percutaneous echo-guided femoral puncture and closure (Figure 1). However, it is important to highlight that this TAVI “simplification” tendency must be accompanied by a rigorous selection of patients who can benefit from a minimalist procedure in order to guarantee procedural safety.

### 2.1. Minimizing Multiple Components of Care

The care pathway for patients undergoing TAVI was initially based on open-surgery standards and often included GA, peri-procedural invasive monitoring, systematic TOE assessment, and intensive care unit (ICU) admission [3].

In 2016, the Vancouver group was one of the first to suggest a new TAVI standardized clinical pathway known as the Vancouver transcatheter aortic valve replacement clinical pathway, which adopts a comprehensive approach for all the TAVI care aspects based on a risk-stratified minimalist peri-procedure approach; standardized post-procedure care, with early mobilization and reconditioning; and criteria-driven discharge home. The main objectives of this initiative were to reduce the heterogeneity in care, identify a subgroup of patients suitable for early discharge (≤48 h), and decrease the length of stay. The new minimalist approach included percutaneous transfemoral vascular access puncture and closure, avoidance of urinary catheter, and development of a LA/CS protocol or extubation in the operative room. The use of other invasive equipment, such as a central venous catheter, a femoral venous sheath for temporary pacing insertion, and a radial artery line, was determined by the anesthesiologist based on the patient’s profile and practice preference. For procedures performed without TOE, TTE imaging was acquired before the vascular access and after the device deployment or within the next 12 h after the procedure. The authors reported that early discharge was associated with LA, implantation of balloon-expandable devices, avoidance of a urinary catheter, and early removal of the temporary pacemaker. They also identified that the challenges associated with implementing the TAVI clinical pathway were related to the program’s historical development and practices, the complexity of the multidisciplinary stakeholder engagement, and the adoption of the length of stay as a quality indicator [4].

Almost at the same time, TAVI groups from different institutions began to suggest the use of enhanced recovery after surgery (ERAS) protocols. The assumption that ERAS clinical pathways could be effectively employed in the TAVI context came from observing the success achieved in several surgical fields. Briefly, ERAS pathways are led by a dedicated heart team and are used as a framework for comprehensive multidisciplinary care, including evidence-based and best practice recommendations for pre-, intra-, and post-operative management. Its implementation has been associated with a reduction in the risk of significant post-operative delirium, cognitive dysfunction, pneumonia, deep venous thrombosis, urinary tract infection, and functional decline. An example of the ERAS pathway was the one suggested by Sola M et al. According to this protocol, the pre-operative phase is centered on nutrition, mobilization, physical therapy, and enhancement of respiratory function. Intra-operative management consists of short-acting anesthetic drugs employment to decrease the likelihood of delirium and allow early functional recovery. Post-operative care focuses on early ambulation, removal of hemodynamic monitors as soon as possible, nutrition, bowel/urinary function evaluation, and opioid-sparing analgesia [5].

The Mack MJ group also suggested a transfemoral-TAVI ERAS protocol based on 6 components: (1) preference for moderate sedation and no intubation; (2) no pulmonary artery or urinary catheters; (3) arterial line removal within 4 h; (4) no post-operative narcotics; (5) mobilization at 4 h and ambulation within 8 h; and (6) antihypertensive reinstitution without nodal blockers. Notably, the authors mentioned that, while ERAS programs are similar in many ways to “fast-track” or “minimalist” protocols, the former encompasses a comprehensive management approach rather than the implementation of one or two “fast-track” items individually. Furthermore, it is intended to be applied to all TAVI patients, leaving room for workflow deviation only when clinically indicated [6].

A few years later, the Vancouver group published the results of the so-called Vancouver 3M protocol (multidisciplinary, multimodality, minimalist TAVI). This protocol includes same-day admission, avoidance of invasive lines (e.g., central venous and urinary catheters), LA only or light procedural sedation administered by an anesthesiologist, percutaneous access and closure, removal of the temporary pacemaker at the end of the procedure, rapid reconditioning, active mobilization after 4 h, accelerated return to baseline functions and daily living activities. Of 1400 patients with severe symptomatic aortic valve stenosis and increased surgical risk screened from low- (<100 TAVI/year), medium-, and high-volume (> 200 TAVI/year) TAVI centers in Canada and the United States, 411 were enrolled. Patients were excluded only if they presented unfavorable vascular anatomy for a percutaneous approach, poor social support, unfavorable airway for emergent intubation, or if they were unable to lie supine. Safe next-day discharge home was achieved in 80.1% of the patients, and within 48 h in 89.5%. The composite outcome of 30-day all-cause mortality or stroke occurred in 2.9% (95% confidence interval (CI) 1.7–5.1%), and it was not affected by the hospital TAVI volume (*p* = 0.51). At 30 days, major vascular complications occurred in 2.4% of the patients, and the cardiac readmission rate was 5.7% [7,8]. Figure 2 illustrates the 3M TAVI clinical pathway.

Similar to the aforementioned, the OCEAN-TAVI registry described the Japanese experience with a minimalistic TAVI approach. The authors enrolled 921 consecutive patients submitted to a transfemoral-TAVI using a minimalist strategy(*n* = 118), which consisted of LA and minimal CS, percutaneous access, and TTE assessment, versus a standard strategy (*n* = 802), which consisted of GA, TOE guidance, with or without bladder catheterization. After propensity score matching, 118 matched pairs were analyzed. Although there was no difference in the rates of in-hospital mortality (2.5% vs. 0.8%, *p* = 0.3) and stroke/transient ischemic attack (1.7% vs. 0.8%, *p* = 0.6), the rates of major or life-threatening bleeding (3.4% vs. 17%, *p* = 0.003) and transfusion (6.8% vs. 29%, *p* = 0.0002) were significantly lower in the minimalistic group. The total ICU days and length of hospital stay were also significantly lower with the minimalistic approach (*p* ≤ 0.0002) [9].

### 2.2. Avoidance of General Anesthesia

TAVI performed under GA presents some potential advantages when compared with LA such as the possibility to use TOE guidance, the opportunity to offer a stable condition during valve deployment, and/or a quickly convert to a bailout surgery in patients with a critical condition. However, GA can, itself, be associated with an increased need for intravenous volume infusion or inotropes drugs to compensate for the for the hemodynamic compromise related to GA induction; in addition, it is frequently associated with higher rates of post-procedural acute kidney injury, pulmonary and renal complications, and longer post-operative length of stay, with additional costs and patient dissatisfaction [9].

A recent analysis from the STS/ACC TVT registry showed a temporal trend toward the use of CS instead of GA. Among 120,080 patients submitted to a transfemoral TAVI from January 2016 to March 2019, the proportion of CS increased from 33% to 63%. The proportion of centers using CS also went from 50% to 76%. Withal, CS was associated with a decrease in in-hospital mortality (adjusted risk difference: 0.2%; *p* = 0.010) and 30-day mortality (adjusted risk difference: 0.5%; *p* < 0.001), a shorter length of hospital stay (adjusted difference: 0.8 days; *p* < 0.001), and a higher chance of being discharged home (adjusted risk difference: 2.8%; *p* < 0.001) when compared with GA [10].

A meta-analysis including 16,543 patients from the German Aortic Valve Registry (GARY) who underwent a TAVI procedure from 2011 to 2014 compared the results of LA or CS versus GA. In the propensity-matched population analysis, patients who received LA or CS presented lower rates of in-hospital low-output syndrome, respiratory failure, delirium, cardiopulmonary resuscitation, and death; shorter ICU stay (odds ratio (OR) 0.82, 95% CI 0.73–0.92; *p* = 0.001), and lower 30-day mortality (2.8% vs. 4.6%; hazard ratio (HR) 0.6; 95% CI 0.45–0.8; *p* < 0.001). Nevertheless, there was no difference in II+ paravalvular leakage incidence (3.9% vs. 4.9%, *p* = 0.13) and 1-year mortality (14.1% vs. 15.5%; HR 0.90; 95% CI 0.78–1.03; *p* = 0.130) [11].

The multicenter randomized SOLVE-TAVI trial evaluated the use of LA with CS versus GA in 447 patients with aortic stenosis submitted to a transfemoral TAVI [12]. For the combined 1-year outcome of all-cause mortality, stroke, myocardial infarction, and acute kidney injury, the authors identified no significant difference between GA (*n* = 61, 25.7%) and CS (*n* = 54, 23.8%; HR 1.09; 95% CI 0.76–1.57; *p* = 0.63). Interestingly, the survival curves started to separate approximately after 6 months in favor of the CS group. Therefore, CS was related to a significantly lower rate of the composite endpoint in a landmark analysis of events occurring between the 30-day and 1-year (GA group: 22.0% vs. CS group: 14.6%, HR 1.57; 95% CI 1.05–2.35; *p* = 0.027) [13].

Harjai KJ et al. assessed the impact of CS (*n* = 278) versus GA (*n* = 199) on procedural efficiency, long-term safety, and quality of life (QoL) in 477 patients with severe aortic stenosis (mean age 82 years, 50% female, 5.0% STS score). CS was used without conversion to GA in 97% of patients and was associated with shorter length of stay (2 vs. 3 days), higher likelihood of being discharged home (87% vs. 72%, *p* < 0.0001), less use of blood transfusion (10% vs. 22%, *p* = 0.0008) and inotropes (13% vs. 32%, *p* < 0.0001), lower contrast volume (50 vs. 90 mL, *p* < 0.0001), and shorter fluoroscopy time (20 vs. 24 min, *p* < 0.0001). At 30 days, the rates of death or stroke (4.0 vs. 6.5%; *p* = NS), and the safety composite endpoint (death, stroke, transient ischemic attack, myocardial infarction, new dialysis, major vascular complication, major or life-threatening bleeding, new pacemaker) were similar (17.6% vs. 21.1%; *p* = NS). At a median follow-up of 365 days, the survival curves showed a similar incidence of death or stroke, as well as the safety composite endpoint between the groups. QoL scores were similar at the baseline and 1 month after TAVI. In multivariable analyses, minimalist-TAVI showed significant improvements in all the parameters of procedural efficiency [14].

The Cribier group described their experience with TAVI over the past two decades and the changes they had observed. One important observation was that, while pre-dilatation was performed almost systematically before 2009 (93%), it was rarely performed after 2014 (14%; *p* < 0.001). The length of stay decreased considerably, with a median duration of only 2 days, and with more than 70% of patients being discharged home within 72 h. They also noticed that transfemoral TAVIs are nowadays performed under LA, with no anesthesiologist in the procedural room, and using exclusive X-ray guidance. Despite these “simplifications”, the observed 30-day mortality rate was as low as 1.4% in 2021, compared with 17% in the initial study period [15].

The topic of TAVI with no anesthesiologist in the procedural room is highly debatable. Denimal T et al. compared TAVI under LA, mostly administered and monitored by a dedicated anesthesia team (regular approach; *n* = 221) versus a new standardized pathway of care, where eligible patients were selected for a minimalist transfemoral TAVI entirely managed by operators, like most interventional coronary procedures (‘‘percutaneous coronary intervention-like’’ approach; *n* = 137). The results showed no differences in the composite safety endpoint according to the VARC 2 definition (Valve Academic Research Consortium–2) (7.3% vs. 11.3%; OR 0.63, 95% CI 0.37–1.07; *p* = 0.086) or in the composite efficacy endpoint (4.4% vs. 6.3%; OR 0.78, 95% CI 0.41–1.49; *p* = 0.45). Patients in the “PCI-like” group experienced a shorter length of stay than those in the regular approach group (5.0 days vs. 6.0 days; *p* = 0.002). In the standardized approach, all vascular accesses were echo-guided, transoperative echocardiography was not used, and unnecessary catheters were avoided. The operator administered the LA with a subcutaneous injection of lidocaine (20 mg/mL). One scrub nurse was assigned to valve preparation. Fluids and drug administration were under the supervision of the implanting physician; and the monitoring of vital signs, level of consciousness, oxygen saturation, and pain score were carried out by another nurse. The authors observed that 75% of the transfemoral TAVI screened patients were eligible for the “PCI-like” strategy. Notably, the main reasons for ineligibility were related to the early experience of the operator with a new valve or device, and patients’ musculoskeletal disorders causing an impossibility to lie supine. They also pointed out that LA was not considered in the presence of any patient-specific or operator-specific clinical reason to prefer GA (TOE required due to the absence of a CT scan or a high risk of paravalvular regurgitation, patient–prosthesis mismatch, or conversion to surgical vascular access), or because of patients’ preference [16]. The observation that a quarter of patients were ineligible to receive the “PCI-like” approach, even in an experienced, high-volume center, makes it clear that TAVI “simplification” cannot be applied to all patients in all settings. Figure 3 presents the “PCI-like” approach eligibility criteria.

Lastly, the use of CS instead of GA frequently implies replacing TOE guidance by TTE assessment or even exclusive X-ray guidance. Despite this, no increase in the incidence of paravalvular leak was observed in current TAVI trials in which TOE guidance was not routinely employed [17].

### 2.3. Vascular Access

Since the PARTNER II trial, the transfemoral TAVI approach has shown better outcomes than alternative access and it is considered the main route for TAVI procedures [18]. Regarding the way to obtain the femoral access, the percutaneous strategy can be performed with a similar or even lower risk of vascular complications and shorter post-procedural length of stay than the surgical cutdown [19]. 

The percutaneous femoral vascular access closure can be performed using either suture or collagen-based devices. The CHOICE-CLOSURE trial investigated which strategy, a pure plug-based (MANTA, Teleflex) or a primary suture-based (ProGlide, Abbott Vascular), would have better outcomes. For this purpose, 516 patients were randomly assigned (mean age 80.5 ± 6.1 years, 55.4% male, 7.6% peripheral vascular disease, 4.1 ± 2.9% mean STS score) to receive one of these two strategies. Although there was no difference in the rate of access site- or access-related bleeding (11.6% vs. 7.4%, relative risk (RR) 1.58, 95% CI 0.91–2.73; *p* = 0.133), device failure (4.7% vs. 5.4%, RR 0.86, 95% CI 0.40–1.82; *p* = 0.841), and length of stay (4.7 ± 2.9 days vs. 4.6 ± 2.4 days; *p* = 0.710), the rate of VARC-2 access site- or access-related major and minor vascular complications during the index hospitalization was significantly lower in the ProGlide group (19.4% vs. 12.0%, RR 1.61, 95% CI 1.07–2.44; *p* = 0.029) [20].

In some hostile femoral accesses, one resource that has been recently employed is the intravascular lithotripsy (IVL). IVL uses circumferential pulse sonic pressure waves to modify vessel intimal and medial calcifications. Due to its potential to treat calcified stenotic iliofemoral arteries, IVL can be used to allow transfemoral access in patients who otherwise would need and alternative access [21]. A European observational multicenter registry reported that IVL was associated with a high procedural success rate (98.2%) and a low rate of complications (1 perforation and 3 major dissections requiring stent implantation) [22].

As relevant as obtaining a safe transfemoral access is identifying which patients are at a high risk of vascular complications. This assessment should be based on pre- (advanced age, female gender, small body surface area, diabetes, congestive heart failure, renal failure, peripheral arterial disease, femoral artery with severe calcification, and small caliber), trans- (use of large-bore sheaths, multiple arterial punctures, no use of ultrasound (US) and fluoroscopy to guide the puncture, no use of micropuncture needle, and inadequate procedural anticoagulation), and post-operative factors (late complications recognition and need for reintervention) [23].

The OxTAVI study identified that the use of US to guide the vascular access, the percutaneous closure, and the active clotting time monitoring were associated with predictable hemostasis and a significant reduction in vascular injury. US-guided vascular access was independently associated with a threefold reduction in vascular access complication (OR 0.29, 95% CI 0.15–0.55, *p* < 0.001). The authors highlighted that US guidance enables direct identification of an anatomically appropriate, calcium-free puncture zone, and subsequent visualization of the puncture; thus, minimizing inadvertent minor vascular injury during the puncture or due to the closure device failure [24].

#### Radial Artery as the Secondary Access or Unilateral Femoral Access

Regarding the second arterial access, while for many years the contralateral femoral artery was the preferable site, a recent study suggested that unilateral femoral access can be done with similar results than bilateral femoral access. For this analysis, 1208 patients (83.36% bilateral vs. 16.64% unilateral femoral access) from the Cleveland Clinic Aortic Valve Center TAVR database were enrolled and analyzed. Meanwhile the use of the unilateral access trended upward significantly over the study duration, reaching 43.7% in 2017, the site-related vascular complication rates declined from 13.7% in 2014 to 7.4% in 2017. After propensity-score matching, peripheral vascular complications were similar between bilateral versus unilateral access (10.8% vs. 8.6%; *p* = 0.543) [25].

Another arterial access often used as a second access site is the radial artery. The radial artery access may decrease TAVI-related vascular complications, improve patient comfort, and allow earlier ambulation [26]. Evaluating the systematic use of the distal radial approach in 41 patients (mean age 76 ± 11.2 years, 41% male sex) undergoing TAVI, Achim A et al. reported no complications related to the transradial access, with a 100% puncture success (defined as completed sheath placement) and only one case of conversion to transfemoral access. On the other hand, transfemoral access complications occurred in 7 cases (17%), with 4 (13.63%) of them been resolved through the distal radial access [27]. Furthermore, many centers have adopted the strategy of using the right radial artery as an additional arterial access for insertion of cerebral embolic protection devices when performing a protected TAVI procedure.

### 2.4. Temporary Pacemaker Approach and Pos-Procedural Conduction Disturbances Management

Rapid pacing is required during balloon-expandable valves deployment or during pre- or post-dilation. Conventionally, the temporary pacemaker is performed via a temporary pacing lead placed in the right ventricle (RV) through a venous access (femoral, jugular, or subclavian vein access). Although this procedure is overall safe, it requires an additional venous access, with the inherent risk of vascular complications, including bleeding, pseudoaneurysm, arteriovenous fistula, thrombosis, or infection. In addition, the temporary pacing lead carries the risk of RV perforation, with pericardial effusion or life-threatening cardiac tamponade, particularly in elderly patients. The use of RV lead also increases procedural and fluoroscopy duration, and procedural costs. To reduce these potential complications, the pacing over the wire technique, which consists of left ventricle (LV) stimulation through the stiff guidewire, has been suggested [24,28].

In terms of procedural efficiency, previous studies reported that pacing over the wire can be achievable in 89–92% of TAVI cases; meanwhile, post-deployment temporary pacing is required in only 4.5–10.6% of cases [28,29,30]. The Ox-TAVI registry demonstrated that the adoption of the LV–guidewire pacing significantly decreased the risk of cardiac tamponade/pericardial effusion (OR 0.19, 95% CI 0.05–0.66, *p* = 0.009) [24].

The EASY TAVI study, a prospective, multicenter, randomized controlled trial, enrolled 303 patients who underwent a transfemoral TAVI with a SAPIEN valve (Edwards Lifesciences, Irvine, California) using LV (*n* = 151) or RV (*n* = 152) stimulation. In the LV group, rapid ventricular pacing was provided via the 0.035-inch stiff guidewire placed in the LV. The cathode of an external pacemaker was attached to the stiff guidewire using a crocodile clip. The anode was attached to the incised skin at the insertion site of the arterial sheath in the groin. Mean procedural duration (48.4 ± 16.9 min vs. 55.6 ± 26.9 min; *p* = 0.0013), fluoroscopy time (13.48 ± 5.98 vs. 14.60 ± 5.59; *p* = 0.02), and cost (€18,807 ± 1318 vs. €19,437 ± 2318; *p* = 0.001) were significantly lower in the LV stimulation group. There was no difference in terms of effective stimulation (84.9% vs. 87.1%; *p* = 0.60) or procedural success rate (100% vs. 99.3%; *p* = 0.99). The 30-day major adverse cardiovascular events rate was 13.9% vs. 17.1% (*p* = 0.44). Therefore, pacing over the wire was associated with significant reductions in procedure duration, fluoroscopy time, and cost, with similar efficacy and safety [28]. Despite these results, EASY TAVI authors pointed out that preventive femoral vein puncture may be required in the presence of high-degree conduction disturbance before TAVI (bifascicular block) to allow immediate temporary pacing after the removal of the LV stiff guidewire [28]. Finally, temporary pacing through external electrode pads may represent a safe rescue alternative when emergency pacing is required [31].

In the technological innovations field, a new TAVI wire (SavvyWire, Opsens Medical) was designed to support transcatheter heart valve (THV) deployment, pacemaker stimulation, and continuous hemodynamic pressure measurement. Although not yet commercially approved, the initial clinical experience with the first 20 patients showed no wire-related adverse events. Appropriate LV pacing (180–220 bpm) and proper systolic blood pressure reduction (reduction of at least 50% resulting in systolic blood pressure < 60 mmHg) were achieved in all the patients. There were no cases of pacing capture failure, and the THV systems were adequately deployed in all cases [32].

Concerning post-operative conduction disturbance management, some authors suggested that intra-operative tests, such as right atrial pacing, could help in identifying patients at increased risk of conduction abnormalities. Briefly, rapid atrial pacing is performed at the end of the procedure by withdrawing the RV pacing wire; then, the atrium is pacing at 10 beats/min increments from 70 to 120 beats/min, for a total of 20 beats at each increment. This strategy aims to evaluate the development of Wenckebach heart block. Patients who develop Wenckebach, especially at lower rates, or those for whom rapid atrial pacing is not feasible (e.g., pre-existing atrial fibrillation) may require an individualized approach. On the other hand, patients at a low risk of additional conduction disturbances can be safely enrolled in an early discharge protocol [33]. For patients who develop a new left bundle branch block with QRS >150ms or PR >240ms with no further prolongation during >48 h after TAVI; and for those with a pre-existing conduction abnormality who develop prolongation of QRS or PR >20ms, the current European guideline suggests that ambulatory electrocardiogram monitoring or electrophysiology study should be considered [34]. Although useful, this strategy may not be widely available, especially in centers with limited resources.

### 2.5. Shorter Length of Stay and Early Hospital Discharge

Some of the factors involved in an early or delayed hospital discharge are the patient clinical condition and comorbidities as age-related functional decline and deconditioning, exacerbated pre-existing disabilities, besides procedure-related factors, hospital-acquired complications, cultural expectations, and family support.

Shorter length of stay offers opportunities for TAVI programs to curb costs, increase capacity, and improve access to care and outcomes. A retrospective cohort study of nearly 15,000 fee-for-service Medicare beneficiaries who underwent elective, uncomplicated transfemoral TAVI in 2016, showed that there was significant heterogeneity across US centers in terms of hospital length of stay, ranging from 1-2 days (49.8%) to 4 days and longer (26.8%). The adjusted cost for next-day discharge was nearly $7500 lower compared with non-next-day discharge (*p* < 0.001) and $5200 lower when controlling for hospital fixed effects (*p* < 0.001) [35]. In addition, a meta-analysis of 8 studies including 1775 participants showed that early discharge (≤3 days after TAVI) was associated with less hospital readmissions [36].

The FAST-TAVI registry validated the appropriateness of a pre-specified set of risk criteria to allow safe and timely discharge. The FAST strategy consisted of LA with CS, US-guided transfemoral puncture, radial approach for secondary arterial access, and LV pacing. Patients were sorted according to the initial strategy (FAST vs. standard). A complete FAST strategy was feasible in 83.0% of cases and all the FAST procedures were successful. The primary outcome of early-safety (all-cause mortality, stroke, life-threatening bleeding, acute kidney injury, coronary artery obstruction, major vascular complication, and valve-related dysfunction) was significantly lower in the FAST group (1.3% vs. 14.3%; *p* < 0.001). The use of the FAST protocol resulted in a reduction of major bleeding (1.3% vs. 10.1%; *p* = 0.01), blood transfusion (2.6% vs. 14.3%; *p* < 0.01), and vascular complications related to the secondary access (0.0% vs. 5.3%; *p* = 0.04). The length of stay was also significantly lower in the FAST group (4.9 days vs. 6.4 days; *p* < 0.01) [37].

In this same line, the multicenter European FAST-TAVI trial presented the experience of multiple European countries (5 in Italy, 2 in the Netherlands, and 3 in the UK) in applying a standardized set of risk criteria to allow timely discharge after TAVI. The pre-specified criteria to define a low risk of complications after early discharge were: NYHA Class ≤II; no chest pain attributable to cardiac ischemia; no untreated major arrhythmias; patients having complications on day 0 to 1, but free of signs or symptoms on day 3; no fever during the last 24 h and no signs of an infectious cause; independent mobilization and self-caring; preserved diuresis (>40 mL/h during the preceding 24 h); no unresolved acute kidney injury type 3; no red blood cell transfusion during the preceding 72 h; stable hemoglobin in two consecutive samples; no paravalvular leak with aortic regurgitation less than moderate; no stroke/transient ischemic attack; and no hemodynamic instability. Among the 502 patients enrolled, 71.0% of those who met low-risk criteria for complications after discharge were actually discharged early. Patients appropriately discharged early presented a significantly lower risk of the combined primary endpoint (all-cause mortality, vascular access-related complications, permanent pacemaker implantation, stroke, re-hospitalization due to cardiac reasons, kidney failure, and major bleeding at 30 days) (7.0 vs. 26.4%; *p* < 0.001), as well as stroke (0.0 vs. 2.8%; *p* = 0.015), permanent pacemaker (4.3 vs. 15.9%; *p* < 0.001), major vascular complications (0.3 vs. 4.7%; *p* = 0.004), and major/life-threatening bleeding (0.3 vs. 6.5%; *p* < 0.001). The authors emphasized that length of stay optimization must be a balance between the potential benefits of early discharge (reduced hospital-borne complications, accelerated patient recovery and mobilization, lower costs) and the potential benefits of late discharge (timely detection and appropriate treatment of infrequent complications such as arrhythmia, as well as vascular and bleeding complications) [38].

Likewise, the results of early (within 24 h after TAVI) versus later discharge were evaluated by Costa G et al. According to this analysis, after adjustment, there were no differences in 30-day all-cause mortality (1.2% vs. 0.0%) or permanent pacemaker implantation (0.6% vs. 0.6%), as well as 1-year all-cause mortality and heart failure re-hospitalization (91.9% vs. 90.6%; *p* = 0.69). Prior permanent pacemaker implantation (OR 2.06, 95% CI 1.21–3.51; *p* < 0.01) and availability of pre-procedural computed CT (OR 1.71, 95% CI 1.15–2.54; *p* < 0.01) were found to be predictors of next-day discharge after TAVI [39].

It should be commented that logistic or social reasons remain the main barriers to early discharge, particularly in older and frail patients. Besides, the objective of a shorter hospitalization length must not be obtained at the expense of safety. Hence, carefully selecting patients who may benefit from an early discharge strategy is the key [6].

## 3. Authors Daily Clinical Practice

The authors’ contemporary minimalist TAVI approach is shown in Figure 4.

In terms of TAVI indications, we follow the current guidelines that recommend TAVI in elderly patients, in patients at a high surgical risk, or in those with comorbidities who would benefit the most from this procedure [40]. All patients are evaluated by a multidisciplinary, collaborative heart valve team, which evaluates clinical, anatomical, and nonclinical (such as psychosocial) factors that could affect the procedural recovery. A pre-operative CT scan (or 3D echocardiography if a CT scan is not suitable) is assessed to estimate the suitability of femoral access, aortic annulus parameters, and potential procedural risk factors. The procedure is usually conducted in a hybrid room. Standard cases are performed under LA and CS. GA is reserved for complex cases when we anticipate a high risk of complications (e.g., a high risk of coronary obstruction or need for a cutdown femoral access). In these cases, an attempt is made to extubate the patient in the operating room, immediately after the procedure. Regardless of the anesthetic approach taken, an anesthesiologist is present in all procedures. The femoral artery is percutaneously accessed and closed, and US is used to guide the punctures. After the femoral artery puncture, the patient is fully heparinized, while at the end of the procedure, the heparin effect is usually reversed with protamine administration. We have adopted the pacing over the wire technique. However, if we anticipate a high risk of conduction disturbances (e.g., pre-existing right bundle branch block), a temporary pacemaker lead is inserted; if no further disturbances occur, it is removed at the end of the procedure. To safely apply this strategy, we carefully test, before inserting the THV, whether the stimulation obtained by the LV stiff guidewire is adequate, without any pacing capture failure. If, post-operatively, a permanent pacemaker is indicated, we follow the current evidence and implant the device within the first 24 h, avoiding any further delay in hospital discharge [33,41]. We do not routinely insert a central venous line or a urinary catheter, or use TOE. At the end of the procedure or within the next 24 h, a TTE is performed to provide post-TAVI assessment. Pre-dilatation is performed only if highly recommended by the bioprosthesis manufacturer; or in the presence of a severe valve calcification, a bicuspid aortic valve, or an aortic valve area < 0.4 cm^2^. Following a standard procedure, active mobilization after 4 h is prioritized, and patients are discharged home in the next 24–48 h. In our experience, early mobilization contributes to promoting a rapid return to baseline status, minimizing nosocomial complications, and decreasing length of stay.

Additional current practices include implanting THV, especially the self-expanding Evolut and Portico platforms, using a cusp overlap fluoroscopy projection (isolating the non-coronary cusp, while overlapping the right and the left coronary cusps), which usually means a right anterior oblique (RAO) and caudal (CAUD) projection. This technique aims to display an elongated view of the aortic root and the LV outflow tract, and allows a precise depth of implantation, reducing the risk of conduction disturbances and the pacemaker rates [42]. This implantation view also favors commissure alignment as the commissure between the right and the left coronary cusps is positioned isolated on the right side of the image. By knowing the location of one commissure, the operators might orient the delivery system so that the neo-commissures are oriented closer to the native ones. In order to facilitate neo-commissure alignment, it has been suggested to insert the delivery system of the Evolut platform with the flush port positioned rotated from a 0 to a 3 o’clock position, the flush port of the Portico/Navitor platform rotated from a 3 to a 6 o’clock position [43], and with the safety button of the Acurate Neo 2 facing down (6 o’clock position).

Furthermore, with TAVI indications expanding to younger patients with longer life expectancy, the concept of lifetime management should be kept in mind when the operators decide which prosthesis best suits which patient, considering anatomical aspects, clinical features, and prosthesis characteristics. Therefore, when we choose the first TAVI prosthesis, we must foresee the possibility of bioprosthesis structural degeneration and a second valve implant, with the subsequent risks of coronary obstruction, patient–prosthesis mismatch, and a residual paravalvular leak.

Finally, in light of all the evidence presented, some caveats must be outlined: do not include in a minimalist approach those unstable patients who could not tolerate local anesthesia or conscious sedation; foresee risky situations, such as the high risk of coronary obstruction, vascular access complications, annulus rupture, or valve mispositioning/embolization; carefully evaluate the pre-operative CT scan to anticipate challenging scenarios; predict a high risk of conduction disturbance and the need for a permanent pacemaker; and, most importantly, never compromise the patient safety.

## 4. Conclusions

As the TAVI volume continuous to increase, minimizing procedural complexity and healthcare resource utilization, while maintaining procedural safety and clinical benefits, is essential. The strategy known as minimalist TAVI, which prioritizes LA and CS, percutaneous vascular access puncture and closure, and early hospital discharge, is growing in popularity and has been associated with results that are, at least, non-inferior compared with conventional strategies. However, the feasibility of this minimalistic TAVI approach depends on several factors, including patients’ characteristics, an anesthesia team able to provide adequate monitored anesthesia, implanters trained in percutaneous access and hemostasis, and an imaging team capable of providing satisfactory post-TAVI assessment. Therefore, the entire multidisciplinary TAVI heart team needs to be engaged in the new TAVI protocols and committed to optimizing procedural care and outcomes.

## Figures and Tables

**Figure 1 jcm-12-01342-f001:**
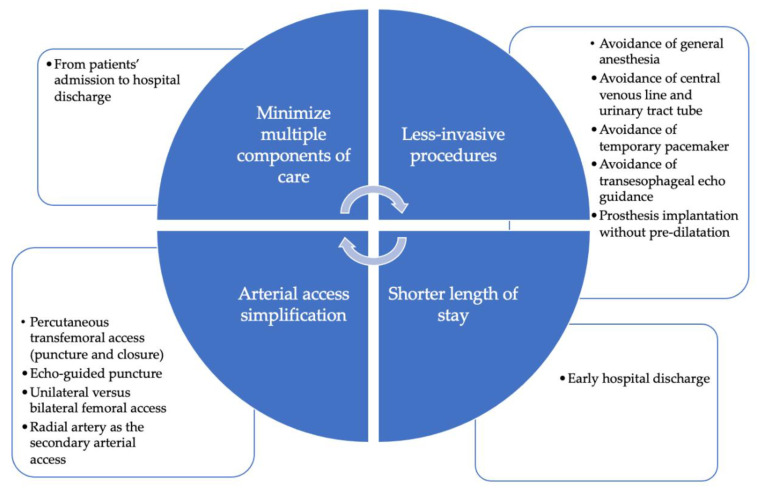
The main steps of a minimalist TAVI approach.

**Figure 2 jcm-12-01342-f002:**
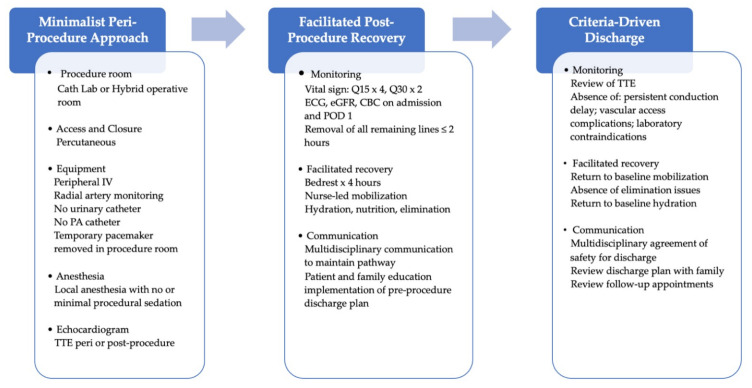
3M TAVI clinical pathway adapted from Wood DA et al. [7]. CBC = complete blood count; ECG = electrocardiogram; eGFR = estimated glomerular filtration rate; IV = intravenous; PA = pulmonary artery; POD1 = post-operative day 1; Q15 = every 15 min; Q30 = every 30 min; TTE = transthoracic echocardiography.

**Figure 3 jcm-12-01342-f003:**
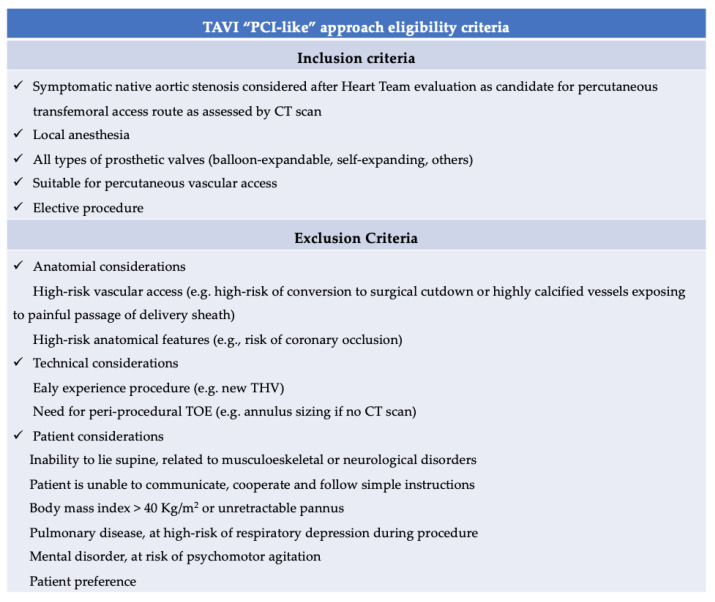
Eligibility criteria for the “PCI-like” approach. Adapted from Denimal T et al. [16].

**Figure 4 jcm-12-01342-f004:**
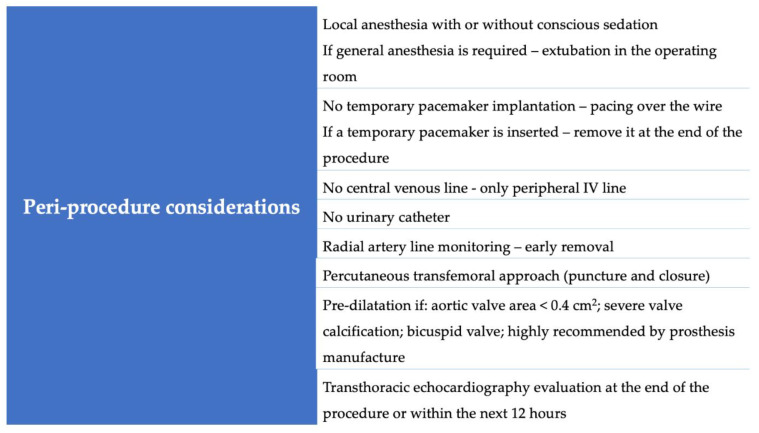
Minimalist TAVI peri-procedure considerations.

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
