# Peer review of "New Practices in Transcatheter Aortic Valve Implantation: How I Do It in 2023"

_jcm, 2023, doi:10.3390/jcm12041342_

Round 1

Reviewer 1 Report

The review titled “New Practices in Transcatheter Aortic Valve Implantation: How I do it in 2023” by Tagliari and Taramasso, is an exhaustive summary of current and new practices in TAVI, particularly concerning the minimalist TAVI approach.

The authors covered all topics regarding the minimalist TAVI approach (type of anesthesia, vascular access, pacing technique during the procedure, etc), reporting data and information from major studies on this field.

The manuscript is well designed and written, only minor spell checks are required.

1) Abstract: “Previously performed under general anesthesia and transoperative transesophageal echocardiography guidance, through a cutdown femoral artery access, the procedure has now evolved into a minimalist approach, where local anesthesia or conscious sedation and avoidance of invasive lines have become the new standards.”

I would modify this sentence as follow: “Previously performed under general anesthesia, with transoperative transesophageal echocardiography guidance, and through a cutdown femoral artery access, the procedure has now evolved into a minimalist approach….”

2) Section 2.2 of the manuscript: “Nonetheless, recently a new TAVI wire designed to allow continuous hemodynamic pressure measurement at the same time that offers support to transcatheter heart valve deployment and pacemaker stimulation was developed”.

This sentence is not flowing, please modify it.

3) Section 3: the number of figure 4 is wrong.

4) Finally, the authors should further elucidate their perspective on this topic, expanding section 3 of the manuscript (Authors daily clinical practice). Describe a step-by-step minimalist TAVI approach from patient selection to patient discharge, according to authors’ clinical practice and summarizing the evidences reported in the manuscript, would be very useful.

Author Response

Dear Editor and Reviewers,

The authors appreciated the reviewers’ comments and suggestions, which will contribute to increasing our article quality and reach. Below, you’ll find the answers to the questions and comments made.

# Reviewer 1.

The review titled “New Practices in Transcatheter Aortic Valve Implantation: How I do it in 2023” by Tagliari and Taramasso, is an exhaustive summary of current and new practices in TAVI, particularly concerning the minimalist TAVI approach.

The authors covered all topics regarding the minimalist TAVI approach (type of anesthesia, vascular access, pacing technique during the procedure, etc), reporting data and information from major studies on this field.

The manuscript is well designed and written, only minor spell checks are required.

1) Abstract: “Previously performed under general anesthesia and transoperative transesophageal echocardiography guidance, through a cutdown femoral artery access, the procedure has now evolved into a minimalist approach, where local anesthesia or conscious sedation and avoidance of invasive lines have become the new standards.”

I would modify this sentence as follow: “Previously performed under general anesthesia, with transoperative transesophageal echocardiography guidance, and through a cutdown femoral artery access, the procedure has now evolved into a minimalist approach….”

Answer: The paragraph has been modified.

2) Section 2.2 of the manuscript: “Nonetheless, recently a new TAVI wire designed to allow continuous hemodynamic pressure measurement at the same time that offers support to transcatheter heart valve deployment and pacemaker stimulation was developed”.

This sentence is not flowing, please modify it.

Answer: The paragraph has been modified.

“In the technological innovations field, a new TAVI wire (SavvyWire, Opsens Medical), designed to support THV deployment and pacemaker stimulation, in addition to allow continuous hemodynamic pressure measurement, was developed. Although not yet commercially approved, initial clinical experience with the first 20 patients was successful, with no guidewire-related adverse effects. Appropriate left ventricular pacing (180-220 bpm) and systolic blood pressure reduction of at least 50% ( resulting in a systolic blood pressure less than 60 mmHg) were achieved in all patients. There were no cases of pacing capture failure, and the THV systems were adequately deployed in all cases [18].”

3) Section 3: the number of figure 4 is wrong.

Answer: The number of the figure has been modified.

4) Finally, the authors should further elucidate their perspective on this topic, expanding section 3 of the manuscript (Authors daily clinical practice). Describe a step-by-step minimalist TAVI approach from patient selection to patient discharge, according to authors’ clinical practice and summarizing the evidences reported in the manuscript, would be very useful.

Answer: A description of our standard TAVI approach has been added.

“The authors’ contemporary minimalist TAVI approach is shown in Figure 4. Following the current guidelines recommendations, TAVI is considered in elderly patients, in patients at high surgical risk, or in those with comorbidities who would benefit more from a less invasive procedure [38]. All patients are evaluated by a multidisciplinary, collaborative heart valve team, which not only participates in the procedural indication but also evaluates clinical, anatomical, and nonclinical (such as psychosocial) factors that could affect post-procedural recovery. Pre-operative CT scan (or 3D echocardiogram, if a CT scan is not suitable) is evaluated to estimate the suitability of femoral access, aortic annulus parameters, and potential procedural risk factors. The procedure is usually executed in a hybrid room. Standard cases are performed with local anesthesia and conscious sedation. General anesthesia is reserved for complex and challenging cases when we anticipated a high risk of complications (e.g., high risk of coronary obstruction or need to cutdown femoral access or alternative access). In these cases, an attempt is made to extubate the patient in the operating room, immediately after the procedure. Regardless of the anesthetic approach taken, an anesthesiologist is present in all procedures. Regarding the pacing strategy, we have adopted the pacing over the wire technique. However, if we anticipated a high risk of conduction disturbances (e.g., pre-existing right bundle branch block), a temporary pacemaker lead is inserted; if no further disturbances occur, it is removed at the end of the procedure. To safely apply this strategy, we carefully test, before the THV insertion, whether the stimulation obtained by the stiff guidewire placed inside the left ventricle is adequate, without any pacing capture failure. If  postoperatively a permanent pacemaker is indicated, we follow current evidence and implant the device within the first 24 hours, avoiding further delay in hospital discharge [39, 40]. We do not routinely insert a central venous line or a urinary catheter or use TOE. At the end of the procedure or maximum within the next 24 hours, a TTE is performed to document the final result. Pre-dilatation is performed only if highly recommended by the bioprosthesis manufacturer or in the presence of severe calcification, a bicuspid aortic valve or an aortic valve area < 0.4 cm2. With this conduct, we avoid unnecessary periods of hypotension, which may increase the risk of calcium embolization, stroke, and myocardial ischemia. Following a standard procedure, active mobilization after 4 hours is prioritized, and patients are discharged home in the next 24-48 hours. In our experience, early mobilization contributes to promoting a rapid return to baseline status, minimizing nosocomial complications, and decreasing length of stay.”

Reviewer 2 Report

Trancatheter aortic valve implantation (TAVI) is one of the most important development in the clinical medicine – cardiology, cardiac surgery and anaesthesiology of the 21th century. The number of these interventions raised dramatically in the last two decades, which induced continuous and rapid development of devices, techniques, indications of TAVI procedures. One of the main directions of this evolution is minimization of the invasiveness of these interventions to reduce patients’ trauma, stress, perioperative morbidity and mortality.

The authors give a detailed up-to-date overview of the main elements of minimalist TAVI approach from adequate patients’ selection and admission through less invasive anaesthesiological  and surgical techniques to shorter hospital stay and earlier discharge. They present step-by-step the most commonly used practical solutions, tips and tricks suggested by the minimalist TAVI protocols. Benefits and results of each intervention, technical and logistical modifications are thoroughly discussed and are backed up by recent studies and publications in details.

Finally, the authors shortly summarize their current minimalist TAVI practice as a synthesis derived from the research and results of the best clinical practice. Moreover, they highlighted some caveats and special issues of minimalist TAVI approach, such as critical application of this technique in high risk patients and difficult anatomical situations, as well as the advantages and technical aspects of cusp overlap technique using different TAVI valves.

The paper analyses an extremely topical issue, it’s very well structured and written to the point.

Author Response

Dear Editor and Reviewers,

The authors appreciated the reviewers’ comments and suggestions, which will contribute to increasing our article quality and reach. Below, you’ll find the answers to the questions and comments made.

# Reviewer 2:

Trancatheter aortic valve implantation (TAVI) is one of the most important development in the clinical medicine – cardiology, cardiac surgery and anaesthesiology of the 21th century. The number of these interventions raised dramatically in the last two decades, which induced continuous and rapid development of devices, techniques, indications of TAVI procedures. One of the main directions of this evolution is minimization of the invasiveness of these interventions to reduce patients’ trauma, stress, perioperative morbidity and mortality.

The authors give a detailed up-to-date overview of the main elements of minimalist TAVI approach from adequate patients’ selection and admission through less invasive anaesthesiological  and surgical techniques to shorter hospital stay and earlier discharge. They present step-by-step the most commonly used practical solutions, tips and tricks suggested by the minimalist TAVI protocols. Benefits and results of each intervention, technical and logistical modifications are thoroughly discussed and are backed up by recent studies and publications in details.

Finally, the authors shortly summarize their current minimalist TAVI practice as a synthesis derived from the research and results of the best clinical practice. Moreover, they highlighted some caveats and special issues of minimalist TAVI approach, such as critical application of this technique in high risk patients and difficult anatomical situations, as well as the advantages and technical aspects of cusp overlap technique using different TAVI valves.

The paper analyses an extremely topical issue, it’s very well structured and written to the point.

Answer: We really appreciate the reviewers' comments.

Reviewer 3 Report

Dear editor,

With interest I read the submitted manuscript entitled “New Practices in Transcatheter Aortic Valve Implantation: How I do it in 2023”.

General comments

In this narrative review the authors discuss the minimalist TAVI approach including their local practice. It is mostly an interventionist’s view; the anaesthesiologist’s view is somewhat lacking.

Although the change from GA to CS plays an important role in the minimalist TAVI approach, the benefits end risks of different sedation techniques vs GA (with orotracheal intubation or with laryngeal mask) are not discussed.

The minimalist TAVI approach corresponds to a worldwide trend (related to the inclusion of low-risk patients, increased experience of the teams, as well as better material and devices with smaller diameters). The simplified procedures are already well described in the literature; we too have a similar approach in our institution (avoidance of central lines and urinary catheters, CS in > 95% of the patients, selective use of temporary PM, in general no pre-dilation, cerebral protection device insertion via the right radial artery).

To make the paper more interesting, the authors should focus mainly on their own experience (“How I do it in 2023”!): patient selection and pathway in their institution, heart-team?, ERAS program underway or planned?, when CS (which drugs ?) when GA, discharge-protocol etc. and present their local approach in the light of the current literature. Now, the paper lacks novelty…

Specific comments:

Temporary pacemakers: have the authors a different approach when using self-expanding valves compared to balloon-expandable valves as the risk of a delayed AVB is higher?

GA vs CS:

As the SOLVE-TAVI trial has shown, using GA has nothing to do with prolonged hospital stay, respectively with worse outcome per se. Also, thousands of patients undergo ambulatory surgery under GA each day without any problems.

The higher renal complication rate, the longer postoperative stay, the increased transfusion rate, as well as the prolonged fluoroscopie time (respectively the higher contrast volume) seen in observational studies are related to patient selection.

The use of CS can also be associated with respiratory depression and therefore be more demanding in high-risk patient with respiratory problems than GA. Knowing the authors local practice would be interesting.

The authors should add the following reference: Thiele H, Kurz T, Feistritzer HJ, et al. General Versus Local Anesthesia With Conscious Sedation in Transcatheter Aortic Valve Implantation: The Randomized SOLVE-TAVI Trial. Circulation. 2020; 142(15):1437-1447

Shorter length of stay and early hospital discharge:

The superior outcome of FAST patients actually seen in observational studies is obviously related to healthy patient selection.

I am however curious if there are any ERAS (enhanced recover after surgery) protocols being evaluated, targeting especially old and frail patients.

Author Response

Dear Editor and Reviewers,

The authors appreciated the reviewers’ comments and suggestions, which will contribute to increasing our article quality and reach. Below, you’ll find the answers to the questions and comments made.

# Reviewer 3:

Dear editor,

With interest I read the submitted manuscript entitled “New Practices in Transcatheter Aortic Valve Implantation: How I do it in 2023”.

General comments 

In this narrative review the authors discuss the minimalist TAVI approach including their local practice. It is mostly an interventionist’s view; the anaesthesiologist’s view is somewhat lacking.

Although the change from GA to CS plays an important role in the minimalist TAVI approach, the benefits end risks of different sedation techniques vs GA (with orotracheal intubation or with laryngeal mask) are not discussed.

The minimalist TAVI approach corresponds to a worldwide trend (related to the inclusion of low-risk patients, increased experience of the teams, as well as better material and devices with smaller diameters). The simplified procedures are already well described in the literature; we too have a similar approach in our institution (avoidance of central lines and urinary catheters, CS in > 95% of the patients, selective use of temporary PM, in general no pre-dilation, cerebral protection device insertion via the right radial artery).

To make the paper more interesting, the authors should focus mainly on their own experience (“How I do it in 2023”!): patient selection and pathway in their institution, heart-team?, ERAS program underway or planned?, when CS (which drugs ?) when GA, discharge-protocol etc. and present their local approach in the light of the current literature. Now, the paper lacks novelty…

Shorter length of stay and early hospital discharge: The superior outcome of FAST patients actually seen in observational studies is obviously related to healthy patient selection. I am however curious if there are any ERAS (enhanced recover after surgery) protocols being evaluated, targeting especially old and frail patients.

Answer: A description of our standard TAVI approach has been added. However, as both authors are surgeons, we feel that we do not have the necessary knowledge to discuss anesthetic topics with adequate depth and robustness. Undoubtedly, this theme deserves a specific article, dedicated to discussing the anesthetic approach during TAVI by specialists. I was wondering if the reviewer could suggest potential authors, who may be invited to delve deeper into the subject and write an article to be part of this same special edition. In addition, a paragraph discussing ERAS protocols has been added.

“The authors’ contemporary minimalist TAVI approach is shown in Figure 4. Following the current guidelines recommendations, TAVI is considered in elderly patients, in patients at high surgical risk, or in those with comorbidities who would benefit more from a less invasive procedure [38]. All patients are evaluated by a multidisciplinary, collaborative heart valve team, which not only participates in the procedural indication but also evaluates clinical, anatomical, and nonclinical (such as psychosocial) factors that could affect post-procedural recovery. Pre-operative CT scan (or 3D echocardiogram, if a CT scan is not suitable) is evaluated to estimate the suitability of femoral access, aortic annulus parameters, and potential procedural risk factors. The procedure is usually executed in a hybrid room. Standard cases are performed with local anesthesia and conscious sedation. General anesthesia is reserved for complex and challenging cases when we anticipated a high risk of complications (e.g., high risk of coronary obstruction or need to cutdown femoral access or alternative access). In these cases, an attempt is made to extubate the patient in the operating room, immediately after the procedure. Regardless of the anesthetic approach taken, an anesthesiologist is present in all procedures. Regarding the pacing strategy, we have adopted the pacing over the wire technique. However, if we anticipated a high risk of conduction disturbances (e.g., pre-existing right bundle branch block), a temporary pacemaker lead is inserted; if no further disturbances occur, it is removed at the end of the procedure. To safely apply this strategy, we carefully test, before the THV insertion, whether the stimulation obtained by the stiff guidewire placed inside the left ventricle is adequate, without any pacing capture failure. If  postoperatively a permanent pacemaker is indicated, we follow current evidence and implant the device within the first 24 hours, avoiding further delay in hospital discharge [39, 40]. We do not routinely insert a central venous line or a urinary catheter or use TOE. At the end of the procedure or maximum within the next 24 hours, a TTE is performed to document the final result. Pre-dilatation is performed only if highly recommended by the bioprosthesis manufacturer or in the presence of severe calcification, a bicuspid aortic valve or an aortic valve area < 0.4 cm2. With this conduct, we avoid unnecessary periods of hypotension, which may increase the risk of calcium embolization, stroke, and myocardial ischemia. Following a standard procedure, active mobilization after 4 hours is prioritized, and patients are discharged home in the next 24-48 hours. In our experience, early mobilization contributes to promoting a rapid return to baseline status, minimizing nosocomial complications, and decreasing length of stay.”

Specific comments:

  • Temporary pacemakers: have the authors a different approach when using self-expanding valves compared to balloon-expandable valves as the risk of a delayed AVB is higher?

Answer: We used the same approach regardless of the TAVI design. Our first-line strategy is the pacing over the wire technique, unless we have identified risk factors for conduction disorders. In this case, especially if we intend to implant an ES, we prefer to use a temporary pacemaker lead in the right ventricle. When we implant BS valves, we carefully check that the pacing over the wire technique is working correctly, with no capture failure. If the pacing is not adequate, we insert the pacemaker lead, but this is extremely unusual if a stiff  guidewire is well-positioned in the left ventricle.

  • GA vs CS: As the SOLVE-TAVI trial has shown, using GA has nothing to do with prolonged hospital stay, respectively with worse outcome per se. Also, thousands of patients undergo ambulatory surgery under GA each day without any problems. The higher renal complication rate, the longer postoperative stay, the increased transfusion rate, as well as the prolonged fluoroscopy time (respectively the higher contrast volume) seen in observational studies are related to patient selection. The use of CS can also be associated with respiratory depression and therefore be more demanding in high-risk patient with respiratory problems than GA. Knowing the authors local practice would be interesting.

 Answer: We use CS as our first-line approach. However, we feel it extremely important to emphasize that our anesthetic team is always present in the operating room and fully capable of converting from one approach to another. The final decision on the anesthetic strategy must be individualized and, if the patient is not considered suitable for CS, GA will be used (e.g., agitated patient, unable to remain in a comfortable resting position during the procedure).

  • The authors should add the following reference: Thiele H, Kurz T, Feistritzer HJ, et al. General Versus Local Anesthesia With Conscious Sedation in Transcatheter Aortic Valve Implantation: The Randomized SOLVE-TAVI Trial. 2020; 142(15):1437-1447

 Answer: The reference has been included.

Reviewer 4 Report

This is a review of the evolution in the TAVI approach with years. It underlines the different components of the hole program, from selection to discharge focusing on the different phase and describing what has changed in the years.

Some comments: The authors should underline that changes like LA vs GA took place in some countries like US in different years in respect to Europe, this is important as the experience is far more dated then the one described in Vancouver experience. The same is for TOE that is more an overseas tradition. Of importance is the paragraph where the authors talk about the PCI like TAVI. the authors should underline that this evolution is not to be interpreted with the equation PCI like= TAVI can be done everywhere. The problem of short length of stay and CD need some further considerations: the relation of PM indication in TAVI are not uniform and either the DRG reimbursement and both influence in hospital stay.

Author Response

Dear Editor and Reviewers,

The authors appreciated the reviewers’ comments and suggestions, which will contribute to increasing our article quality and reach. Below, you’ll find the answers to the questions and comments made.

# Reviewer 4.

This is a review of the evolution in the TAVI approach with years. It underlines the different components of the hole program, from selection to discharge focusing on the different phase and describing what has changed in the years.

Some comments: The authors should underline that changes like LA vs GA took place in some countries like US in different years in respect to Europe, this is important as the experience is far more dated then the one described in Vancouver experience. The same is for TOE that is more an overseas tradition.

Answer: We have added additional protocols and outcomes from different centers around the world to illustrate the employment of the minimalist TAVI approach from different perspectives.

Of importance is the paragraph where the authors talk about the PCI like TAVI. the authors should underline that this evolution is not to be interpreted with the equation PCI like= TAVI can be done everywhere.

Answer: We have added an observation.  

“The observation that a quarter of patients were ineligible to receive the “PCI-like” approach, even in an experienced, high-volume center, makes it clear that TAVI “simplification” cannot be applied to all patients in all settings places.”

The problem of short length of stay and CD need some further considerations: the relation of PM indication in TAVI are not uniform and either the DRG reimbursement and both influence in hospital stay.

Answer: We have included a suggestion on how to manage CD according to the recommendations expressed by recent guidelines.

Reviewer 5 Report

This is a very well written article on minimalistic approach for TAVR. This is the need of the hour to decrease hospital stay without increasing risk for patients. There are minor points that can be added:

1. There are some patients who will develop complete heart block or require pacemaker implantation. What methods are suggested by authors to identify these patients if we are going for early discharge? There are reports of certain strategies that are being used which include implantable loop recorders/ambulatory ECG monitoring (PMID: 30031719, 36362484) or identification of high risk patient during procedural rapid pacing (33478630). Can this be incorporated? Why or why not?

2. What suggestions do authors have for patients with non favorable transfemoral access. There have been reports of use of intravascular lithotripsy to make femoral access accessible (PMID: 33358548, 34631837). Can this play a role decreasing length of stay (LOS) for patients who need alternative access? 

3. What is the use of vascular closure devices to prevent vascular complications and arterial access simplification? Is there a role of these devices in reducing the LOS. Is there a difference in type of device used and outcomes, especially in reduction of LOS? (https://doi.org/10.1016/j.jscai.2022.100397). There are preventive measures that can be used to prevent vascular injury, can they help to make TAVR successful with minimalistic approach? (PMID: 33745675)

Overall, this is a very well written article.

Author Response

Dear Editor and Reviewers,

The authors appreciated the reviewers’ comments and suggestions, which will contribute to increasing our article quality and reach. Below, you’ll find the answers to the questions and comments made.

# Reviewer 5.

This is a very well written article on minimalistic approach for TAVR. This is the need of the hour to decrease hospital stay without increasing risk for patients. There are minor points that can be added:

  1. There are some patients who will develop complete heart block or require pacemaker implantation. What methods are suggested by authors to identify these patients if we are going for early discharge? There are reports of certain strategies that are being used which include implantable loop recorders/ambulatory ECG monitoring (PMID: 30031719, 36362484) or identification of high risk patient during procedural rapid pacing (33478630). Can this be incorporated? Why or why not?

Answer: We have added a paragraph discussing this topic.

“Concerning conduction disturbance, some authors have suggested that intra-operative tests, such as right atrial pacing, could help in identifying patients at increased conduction abnormalities risk. Briefly, rapid atrial pacing is performed at the end of the procedure by withdrawing the right ventricular pacing wire and then pacing the atrium at 10 beats/min increments from 70 to 120 beats/min, for a total of 20 beats at each increment. This strategy aims to evaluate the development of Wenckebach heart block. Patients who develop Wenckebach, especially at lower rates, or those for whom rapid atrial pacing is not feasible (e.g., pre-existing atrial fibrillation) may require an individualized approach. On the other hand, patients who are at low risk of additional conduction disturbances can be safely enrolled in an early discharge protocol [41]. For patients who develop with new left bundle branch block with QRS >150ms or PR >240ms with no further prolongation during > 48h after TAVI, and for those with a pre-existing conduction abnormality who develop prolongation of QRS or PR >20ms, the current European guideline suggests that ambulatory electrocardiogram monitoring or electrophysiology study should be considered [39]. While useful, this strategy may not be widely available, especially in centers with limited resources.”

  1. What suggestions do authors have for patients with non favorable transfemoral access. There have been reports of use of intravascular lithotripsy to make femoral access accessible (PMID: 33358548, 34631837). Can this play a role decreasing length of stay (LOS) for patients who need alternative access? 
  2. What is the use of vascular closure devices to prevent vascular complications and arterial access simplification? Is there a role of these devices in reducing the LOS. Is there a difference in type of device used and outcomes, especially in reduction of LOS? (https://doi.org/10.1016/j.jscai.2022.100397). There are preventive measures that can be used to prevent vascular injury, can they help to make TAVR successful with minimalistic approach? (PMID: 33745675)? 

Answer: We have added a paragraph discussing this topic.

“The percutaneous femoral vascular closure can be done either with suture- (e.g., ProGlide) or collagen-based (e.g., or MANTA) devices. The CHOICE-CLOSURE trial investigated which strategy, a pure plug-based (MANTA, Teleflex) or a primary suture-based (ProGlide, Abbott Vascular), would have better outcomes. For this purpose, 516 patients were randomly assigned (mean age 80.5±6.1 years, 55.4% male, 7.6% peripheral vascular disease, 4.1±2.9% mean STS score). Although there was no difference in access site- or access-related bleeding (11.6% vs. 7.4%, relative risk (RR) 1.58, 95% CI 0.91 – 2.73; p=0.133), device failure (4.7% vs. 5.4%, RR 0.86, 95% CI 0.40 – 1.82; p=0.841), and length of stay (4.7±2.9 days vs. 4.6±2.4 days; p=0.710), the rate of VARC-2 access site- or access-related major and minor vascular complications during the index hospitalization was significant lower in the ProGlide group (19.4% vs. 12.0%, RR 1.61, 95% CI 1.07 – 2.44; p=0.029) [21].

In some hostile femoral accesses, one resource that has been recently employed is intravascular lithotripsy (IVL). IVL uses circumferential pulse sonic pressure waves to modify both vessel intimal and medial calcifications and is a potential treatment option for calcified, stenotic iliofemoral artery disease. Among its benefits is the potential to reduce transfemoral-TAVI related vascular complications and minimize the amount of patient requiring alternative non-transfemoral access, thus contributing to keep the feasibility of a minimalistic totally percutaneous transfemoral TAVI approach, even in challenging scenarios [22].  A European observational multicenter registry have reported that IVL was associated with a high procedural success rate (98.2%, 2 patients needed conversions to cardiac open surgery for annular rupture and valve migration) and a low rate of complications related to the IVL-treated segments (1 perforation and 3 major dissections requiring stent implantation) [23]. 

As relevant as obtaining a safe transfemoral access is identifying which patients are at high risk of vascular complications. This assessment should be based on pre- (advanced age, female gender, small body surface area, diabetes, congestive heart failure, renal failure, peripheral arterial disease, femoral artery with severe calcification and small caliber), trans- (use of large-bore sheaths, multiple arterial punctures, failure to use ultrasound and fluoroscopy to guide the puncture, failure to use a micropuncture needle, inadequate procedural anticoagulation) and post-operative factors (signs and symptoms of complications and indication for reintervention) [24].”

Round 2

Reviewer 3 Report

Dear editor,

Thank you for letting me re-review the submitted manuscript entitled “New Practices in Transcatheter Aortic Valve Implantation: How I do it in 2023”.

Here my comments:

The authors made several improvements as including a paragraph regarding the current state of ERAS programs for TAVI patients.

They also explain in more detail their local approach. Their local CS technique remains undescribed as their anesthesiologists are not involved in the paper (I wonder why….)

The discussion of the SavvyWire in the GA vs CS paragraph should probably be placed in the pacemaker paragraph.

Author Response

Dear Reviewer,

Thank you for your comments.
As I mentioned, the explanation is simple. This article was thought to open this Special Issue second edition and, therefore, our intention, as Editors, was to raise interesting and current TAVI topics precisely to encourage more authors to publish new articles on the subject in our Special Issue, and not to delve into each of the topics, under penalty of this article, which is already extremely long, becoming even more longer.

The paragraph has been relocated.

Reviewer 4 Report

The paper has improved and though it is a subject already widely investigated by other authors it gives a nice summary of the various matters.

Author Response

We really appreciate the reviewers' comments.